# Machine-Learning Algorithms for Process Condition Data-Based Inclusion Prediction in Continuous-Casting Process: A Case Study

**DOI:** 10.3390/s23156719

**Published:** 2023-07-27

**Authors:** Yixiang Zhang, Zenggui Gao, Jiachen Sun, Lilan Liu

**Affiliations:** Shanghai Key Laboratory of Intelligent Manufacturing and Robotics, School of Mechatronic Engineering and Automation, Shanghai University, Shanghai 200444, China; archezhang@foxmail.com (Y.Z.); gaozg@shu.edu.cn (Z.G.); jiachensun1997@163.com (J.S.)

**Keywords:** machine learning, continuous casting, inclusions, imbalanced dataset, case study, quality prediction

## Abstract

Quality-related prediction in the continuous-casting process is important for the quality and process control of casting slabs. As intelligent manufacturing technologies continue to evolve, numerous data-driven techniques have been available for industrial applications. This case study was aimed at developing a machine-learning algorithm, capable of predicting slag inclusion defects in continuous-casting slabs, based on process condition sensor data. A large dataset consisting of sensor data from nearly 7300 casting samples has been analyzed, with the empirical mode decomposition (EMD) algorithm utilized to process the multi-modal time series. The following machine-learning algorithms have been examined: K-Nearest neighbors, support vector classifier (linear and nonlinear kernels), decision trees, random forests, AdaBoost, and Artificial Neural Networks. Four over-sampling or under-sampling algorithms have been adopted to solve imbalanced data distribution. In the experiment, the optimized random forest outperformed other machine-learning algorithms in terms of recall and ROC AUC, which could provide valuable insights for quality control.

## 1. Introduction

Monitoring the final properties of steel manufactured products is an indispensable procedure to guarantee their final quality. As one of the key processes in steel production in the modern industry, continuous casting has become a research focus of steel production companies, design, engineering, and research institutes. In the continuous-casting process, liquid (molten) steel is poured from a ladle into the tundish, then flows through submerged entry nozzles into one or several water-cooled copper molds [1]. The copper mold serves as the primary cooling unit, where a solidified shell forms around the molten steel. The cooling process of metal (or casting strand) is finished by several water-spray cooling zones, referred to as secondary cooling [2]. At the end of the casting line, the strand is cut off and transferred for inspection, then into rolling and finishing sequences. The schematic diagram of a continuous-casting machine is presented in Figure 1.

While continuous-casting methods ensure higher productivity of steel, continuous improvement of the product quality has long been a challenging issue, due to the complexity and nonlinearity involved. Therefore, the ability either to reveal the formation of product defects or to predict the quality of continuous-casting products, whether based on simulation or process data analysis, can contribute to the decrease in costs in terms of inspection, production, and management throughout the whole steel-making process. More specifically, modeling and analysis concerning steel cleanliness, which implies a tight control of the inclusion formation, is essential to fulfilling the increasing demands for quality and efficiency.

In the past several decades, advanced modeling and computer simulation techniques have shown their capabilities in knowledge discovery in the continuous-casting process, i.e., explaining how heat transfer and fluid flow in the process would affect the quality of products. Regarding the flow patterns in the continuous-casting mold, Researchers in [3] developed a mathematical model based on an existing three-dimensional flow field model, considering the transport equation of non-metallic inclusions. In “Flow Dynamics and Inclusion Transport in Continuous Casting of Steel [4]”, the results of several model simulations on transient flow patterns in the continuous casting of steel were reported. Sink term and full-solidification methods were applied in 3-dimensional numerical models to predict the distribution of non-metallic particle inclusion in the full length of caster strands. The model predictions showed generally well correspondence with the actual distribution of non-metallic inclusions acquired by industrial measurement [5]. Zhang et al. [6] established a water model of a slab mold to analyze flow-field behaviors, using particle image velocimetry measurements and fluctuation sensors for the analysis of the slag entrainment and inclusion adsorption. Gupta et al. [7] dealt with the development of a multiphase numerical model of the continuous-casting process of steel bloom, considering different submerged entrance nozzle (SEN) configurations that differ in port angles for the study of flow pattern, temperature distribution, solidification thickness, interface fluctuation, and inclusion behavior during the process. Wang et al. [8] established a kinetic model coupled with thermodynamic equilibrium and kinetic diffusion to investigate the formation of inclusion in the tinplate-casting process and the composition transformation of slag inclusions in molten steel in a continuous-casting mold. Some relevant studies also considered the effects of electric magnetic stirring. For instance, Ji et al. [9] investigated the effect of electromagnetic stirring on a continuous-casting strand on the distribution of inclusions along the thickness of gear steel blooms, concluding that unreasonable current electromagnetic stirring operation parameters will lead to slag entrainment. Li et al. [10] established a mathematical model, which coupled the electromagnetic field and fluid-flow field, to clarify the issues regarding the positions and working patterns of electric magnetic stirring devices. The authors suggested that to improve the inclusion removal, the stirrer be installed below the SEN port.

Compared to modeling and simulation methods, which are mostly more simplified, situational-specific solutions [11], data-driven prediction and analysis methods, inspired by the advent of industrial IoT technologies, formed more distinct and effective alternatives for applied process supervision in the continuous-casting process. By mining patterns and trends in the source data, the data-driven approaches have shown their usefulness in exploring the relationships between the steel-fabrication parameters and the presence of defects in the final products, from a statistical perspective. Focusing on conventional machine-learning-based approaches, Matsko et al. [12] provided an outline for the application of an adaptive fuzzy decision tree for the automatic process control of the CCP. Zhao et al. [13] proposed an iPSO-LSSVM-based model for the prediction of the intermixing length in the steel-grade transition casting process. The proposed method reached the lowest prediction errors among linear regression, nonlinear regression, and iPSO-LSSVM methods. Cuartas et al. [14] classified steel castings for tire reinforcement depending on the presence and properties of non-metallic inclusion. The optimized random forest with the highest probability of identifying defective samples and explaining feature importance was selected to improve quality control. Other researchers also focused on the application of deep-learning techniques in the monitoring of the continuous-casting process. Kong et al. [15] developed a prediction model based on a neural network and expert system for internal cranks during slab continuous casting. Using the internal crack generation index of slice units of products, the system reached an overall accuracy of 86.85% in the application of a steel mill. Zhou et al. [16] employed multi-scale CNN-LSTM to detect anomalies in time-dependent continuous-casting process parameters, and fused anomaly detection results to predict the inclusions in steel slabs. Similarly, Wu et al. [17] adopted multi-scale CNN and RNN models for reliable quality prediction of continuous-casting slabs. The authors also conducted data balancing based on the random under-sampling (RUS) method to mitigate the impact of the skewed data distribution.

Apart from quality inspection in the continuous-casting process, other state-of-the-art research has also revealed the potential of machine-learning algorithms in providing precise and interpretable predictions in various applications. Dhaliwal et al. [18] proposed an XGBoost-based model for measuring the parameters of data in networks. Zhang [19] proposed cost-sensitive K-nearest neighbor (KNN) classifiers by minimizing misclassification costs in imbalanced classification processes. Memiş et al. [20] combined soft sets and KNN classifiers, forming a fuzzy parameterized fuzzy soft KNN classifier, which obtained optimal performances in a comprehensive comparison experiment. Also based on fuzzy theories, Kaminska et al. [21] proposed an improved KNN using ordered weighted average operators and obtained optimal results in applied aspect-based sentiment analysis. Kaushik et al. [22] used a multinomial naive Bayesian Classifier for text-classification-based sentiment analysis. Erkan [23] reported an eigenvalue-based algorithm, using the eigenvalues of each sample in the test data concerning the training data to make classifications.

The studies mentioned above, as well as others, have shown the potential of data-driven machine-learning methods in the analysis and estimation of production processes in intelligent manufacturing.

To the best of the authors’ knowledge, only minor attention has been paid to the prediction of the slag inclusion in continuous-casting slabs based on sensor time series and machine-learning algorithms, which could potentially provide a simple yet effective alternative for the knowledge discovery and process control in this specific area. As a result, several well-established and fine-tuned models, as well as a variety of popular time-series-processing, feature-extraction, and data-balancing techniques, have been implemented in this research and validated in order to derive informative predictions. This seems to be a promising approach to reveal the success, as well as the limitations, of machine-learning methods in the time-series-based prediction for the inclusion quality of steels. Based on former studies, the paper introduced the following contributions:Introduces time-series-driven analysis and defect prediction into the continuous-casting process.An EMD algorithm has been implemented to effectively decompose continuous-casting sensor signals and enhance the signal by reconstructing a multi-variate signal using the most informative IMFs.Conducts a comprehensive study on the topics in continuous-casting time-series-based quality prediction, ranging from feature engineering and data balancing to the application of machine-learning models.

The rest of this paper is organized as follows. The data source and feature-engineering processes introduced in this case study, along with machine-learning algorithms are introduced in Section 2. The evaluation metrics and performance of the algorithms are compared through experiments in Section 3. Section 4 concludes this paper and makes recommendations for further studies.

## 2. Materials and Methods

### 2.1. Casting Slabs and Quality-Control Issues

Continuous casting is one of the main procedures employed to obtain steel cords, sheets, or planes. The molten steel from furnaces is transmitted into ladles, undergoes ladle treatments, such as alloying, and after arriving at the optimal temperature, transformed into continuous-casting slabs with certain dimensions (In this case, the dimensions of the casting slab vary from 900 mm × 210 mm × 8000 mm to 1950 mm × 300 mm × 10,000 mm). Then, hot-rolling and cold-rolling procedures are applied to produce wide and thin steel plates. Different types of defects may cause the deterioration of the microstructure and mechanical properties of the cast products [24]. These defects may include surface scratches, central segregations, decarburization of the steel, and, especially, the presence of non-metallic inclusions, such as slags in the tundish and mold.

Mold slag (fluxes) is one of the key elements that contributed to the prevention of premature oxidation and solidification of molten steel during continuous casting, provided lubrication for the relative movement of the mold-primary shell, absorbed other oxides (such as alumina, etc.), and ensured the normal operation of the continuous-casting process [25]. However, in the tundish and the casting mold, the liquid mold slag particles floating on the surface of molten steel are easily introduced into the molten steel, especially in the absence of excessive mold-level fluctuations, leading to final defects in the steel samples [26].

In order to achieve the specified compositional ranges of steel, mainly SS330 and SS400, steel mills at present follow a certain set of protocols in the manufacture of continuous-casting slabs. During casting, the following procedures are commonly monitored [14]:Deoxidation.Composition control of special slags in the tundish and casting mold.The temperature, molten steel flow rate, and level control of the ladle, tundish, and mold.The control of oscillation and electromagnetic stirring of the casting mold.

In addition, steel mills have quality procedures for the control of inclusions. All the defective samples that caused the lockdown of rolling lines would be sent for further examination. The availability of such procedures, together with predictive algorithms, could help to foresee the inclusionary condition as a function of the manufacturing parameters, and thus should allow the final quality of the casting steel plates to be improved.

### 2.2. Data Source

The sensor dataset was from a continuous-casting mold in one slab continuous-casting machine at a local steel industry. In the casting process, a variety of parameters were monitored and controlled via industrial sensor networks, including the fluctuations of the mold top surface, the flow rate of argon gas and molten steel in the SEN port, the oscillation amplitude of the casting mold, the temperature in the tundish, and the flow rates of cooling water from the sides of casting mold. To reduce the transmission cost, all sensor signals were down-sampled to 1 Hz and uploaded to the cloud server, where necessary data would be downloaded for further detailed analysis. In this case study, 6 process parameters, which were considered as key variables by experts on-site, were analyzed, including cast speed (CS, m/min), mold-level fluctuation (MLF, mm), and flow rates of mold cooling water from the four sides of the casting mold (CW 1–4, L/min). Records relevant to the lock-down of rolling production lines caused by slag inclusions were also obtained as quality references, which would be processed into binary labels (“normal”-“defective”) of production quality. According to the production lists and quality labels, approximately 7300 slabs, including 327 defective (“positive”) samples, were investigated. Different condition parameters of 4 representative samples in the dataset were displayed in Table 1.

The original signal dataset contains unreadable observations, missing values, and noises due to interference from various sources, which may mislead the training of machine-learning models, resulting in higher model complexity, and longer training time. Hence, unreadable observations and missing values were filled with the mean values of the features. Outliers were removed from the dataset, and replaced with the median value of a 7-point sliding data window. Finally, the signals were prepared for feature extraction. The time series curves after outlier removal are presented in Figure 2.

All the signal-processing blocks and machine-learning models were established in a Python 3.8.8 environment on a personal computer with 16 GB RAM capacity, driven by an Intel i7-10875H CPU. No GPU computing tools have been adopted at any stage through signal processing, feature engineering, or training and testing of machine-learning algorithms.

### 2.3. Preprocessing and Feature Engineering

#### 2.3.1. Signal Processing

Due to the high-dimensional nature and complexity of the continuous-casting process, process parameters could contain information on various modes with different periods and intervals. For instance, Teshima et al. [27] pointed out that mold-level fluctuations were affected by the combination of both long (the impingement of molten steel upper roll flow on the free surface) and short (mold oscillation) period factors. Thus, it is considered necessary to decompose the raw signals into various time scales or frequency bands. One classic signal-processing technique commonly applied in fault diagnosis and pattern recognition, the empirical-mode decomposition (EMD) technique was selected for the decomposition of multi-variate sensor time series for its self-adaptiveness. The basic concepts of the algorithm are introduced below.

EMD is an effective signal processing technique suitable for denoising and blind source separation of nonlinear and nonstationary time series. The method decomposes the raw time series into multiple intrinsic mode functions (IMFs), which normally satisfy the following constraints:The number of extrema and the number of zero-crossings must either equal or differ at most by one.The mean value of the envelope defined by local maxima and the envelope defined by the local minima is zero [28,29].

The general workflow of EMD is presented in Figure 3.

Different from signal-processing techniques, EMD doesn’t rely heavily on the selection of hyperparameters (such as the selection of wavelet functions for wavelet decomposition), thus it is self-adaptive and flexible for time-series data. In this research, a Python integration of EMD was realized, with the maximum number of IMFs set to 8. The result of the EMD decomposition of the MLF signal from one cast sample is presented in Figure 4.

To save on the computing costs of the feature extraction process, and reduce the overall complexity of the prediction model, the normalized energy of each IMF was calculated, following the workflow below.

The energy of IMF j in the signal i on the whole signal length l can be calculated by the following equation:(1)$$E_{ij}=\int \left|s\right|dt=\sum_{i=0}^{l}{d_{i}}^{2}$$

And by summing up all the energy values in the signal i, the IMF energy of the signal can be determined:(2)$$E_{i}=\sum_{k=0}^{j}E_{ik}$$

Next, the energy values can be normalized to avoid a gap between elements and the normalized energy vector of signal i can be calculated:(3)$$T_{i}=\left[w_{1},w_{2},w_{3},\cdots w_{j}\right]=\left[\frac{E_{i1}}{E_{i}},\frac{E_{i2}}{E_{i}},\frac{E_{i3}}{E_{i}},\cdots \frac{E_{ij}}{E_{i}}\right]$$

The normalized energy values of all IMFs were presented in Figure 5. Based on the results, IMFs with normalized energy values larger than 0.1, were selected for the feature extraction process.

#### 2.3.2. Statistical Feature Extraction and Selection

Feature-engineering procedures were executed using Python toolboxes including SciPy and Tsfresh after the data preprocessing. Nine classical condition parameters were obtained by statistical analysis from all the IMF, trends, and residuals on the time domain, including absolute mean value (T0), root mean square (RMS, marked as T1), root amplitude (T2), standard deviation (T3), skewness (T4), kurtosis (T5), crest factor (T6), coefficient of variation (T7), and maximum deviation (T8). Fast Fourier transform (FFT) was applied for each IMF. From the frequency band, central frequency (F0), mean square for frequency (F1), root mean square for frequency (F2), variance for frequency (F3), and signal energy in the frequency domain (F4) were extracted. Another set of important parameters was acquired from the phase spectrums of the transformed IMF signals, including the absolute mean value (F5), RMS (F6), root amplitude (F7), standard deviation (F8), skewness (F9), kurtosis value (F10), crest factor (F11), and maximum deviation (F12). A total of 446 features from 24 IMFs, 12 trend signals, and 12 residuals were included in the feature dataset. The process of data acquisition, signal processing, and feature extraction is presented in Figure 6.

For most of the ML algorithms, multicollinearity could reduce the performance of a model, leading to a higher standard error and higher model complexity as well as making the research on feature importance harder [14]. For this reason, a Pearson-correlation-based filtering method was conducted in the feature-selection process. If the Pearson correlation coefficient between two features was above 0.85, one of the variables was removed. In the feature-selection process, only 126 out of 446 features were preserved. Most of the features eliminated were the descriptions of detailed modes in CW1-CW4, which were highly corresponded from engineering aspects. Pearson correlation-based heatmaps before and after feature selection are presented in Figure 7, showing the effects of removing highly correlated features. The researchers used 60% of the labeled data for model training and 20% for parameter optimization; 20% was preserved for the testing of the tuned model.

### 2.4. Algorithms for Classification and Data Resampling

#### 2.4.1. Machine-Learning Algorithms

In recent years, various machine-learning packages and toolboxes have been released to handle the complexity and high dimensionality in actual industrial scenarios. Packages such as Scikit-learn, Spark ML-Lib, and Weka have provided highly compact and flexible solutions for building up a custom analysis workflow within programming environments, while tools like KEEL, RapidMiner, Tanagra, and Orange provided nearly end-to-end software applications for developing machine-learning algorithms. Considering package compatibility, deployment complexity, and overall dependability [30], the authors have selected Scikit-learn, a Python machine-learning package under constant development and with good algorithm support, to configure the machine-learning models. The pipeline also used the libraries Numpy and Pandas for reading data inputs, and Matplotlib and Seaborn for visualization and analysis of results. After the data was allocated, preprocessing and feature extraction tasks were carried out, and the selected features were used as input for classifiers. The following classification algorithms have been used in this research:K-Nearest Neighbors (KNN) [31]: The KNN algorithm is one of the most simple and basic supervised learning methods for classification and regression in industry. As the name “K-nearest neighbors” suggests, the algorithm is based on the idea that, if most of the k-nearest neighbors of a sample x, which are selected by a specific distance metric throughout the feature space, belong to a certain class label L, then this sample also belongs to this class label. However, KNNs are subject to some problems such as noises and outliers, class imbalance, dependency on neighborhood parameters, and high computing costs for large datasets.Support Vector Machines (SVM): SVM is a supervised learning method widely applied in classification and regression tasks. By mapping the data into high-dimensional space with kernel functions and constructing proper hyperplanes [32], SVM could effectively handle various classification datasets, even when the data distributions are nonlinear [33]. However, the SVM models are hard to fit on massive data and are sensitive to kernel parameters and normalization strategies. As a result, it might be a suboptimal choice when outliers and missing values are constant in input datasets.Decision Trees (DT): DT models are competitive alternatives for supervised learning tasks such as prediction, forecasting, and statistical feature selection. A DT classification model selects the best attribute using attribute-selecting measures to split records, designates the attribute as a decision node, breaks the dataset into smaller subsets, and recursively iterates the process on the child decision nodes until the stopping criteria are met [33]. DT models are simple and naturally interpreted, but may suffer from high variance and excessive complexity in large high-dimensional datasets.DT ensembles: The ensemble algorithm is based on the concept of combining multiple “weak” classifiers into a “strong” classifier [34]. The most popular ensemble models are arguably those based on DT algorithms, such as ransom forests (RF) and Adaptive Boosting (AdaBoost) models, which would be validated in this case study. In RF models, each DT is built simultaneously based on randomly generated subsets of the dataset. On the other hand, weak trees in AdaBoost are trained sequentially, each one trying to correct its predecessor.Artificial Neural Networks (ANN): ANNs are widely applied in various data-driven classification and forecasting. A typical ANN is a multi-layer, which could map highly complex patterns through multiple nonlinear transformations [35]. The back-propagation neural network (BPNN) is one of the most prevalent ANN models, which was tested and compared in this case study. Given the input/output pairs, BPNN can have its weights adjusted by the back-propagation (BP) algorithm to capture the non-linear relationships in complex datasets [36].

#### 2.4.2. Resampling of Imbalanced Data

The number of “defective” castings in the final dataset was significantly lower than the “Normal” castings, with a rejection rate of only 4.2%. Such highly skewed datasets often pose a challenge for informative predictive modeling. There are different methods for dealing with imbalanced sets, which may be categorized as follows: resampling of training sets, alternative evaluation metrics, and weighting of different classes. In this case study, alternative evaluation metrics (see Section 3.1 Evaluation metrics) and data resampling were considered to handle the imbalanced dataset. In this subsection, the following resampling algorithms were compared using imblearn (version 0.11.0)-based implementations in Python 3.8.8:Random under-sampling (RUS) and random over-sampling (ROS): RUS and ROS are among the simplest algorithms for balancing the data. The RUS method randomly selects major class instances and removes them from the dataset until the desired class distribution is achieved [17], while ROS randomly selects minor class instances and adds their copies into the dataset until the dataset reaches a more balanced distribution.Near-Miss (NM): The Near-Miss under-sampling technique works based on the widely-used nearest-neighbor method. First, the distances between majority and minority instances are calculated. Then, the nearest majority neighbors of the minority instances are selected and retained [37]. In this case study, the 3rd version of the Near-Miss algorithm was selected for estimation.Synthetic Minority Over-sampling Technique (SMOTE): SMOTE is a synthetic over-sampling algorithm that carries out an oversampling approach to rebalance the original dataset [38,39]. Instead of producing merely replicas of minority classes, SMOTE works by iteratively finding the nearest neighbors of the minority instances and generating new minority instances through the computation of distances.

## 3. Results and Discussions

### 3.1. Evaluation Metrics

An accuracy score is one of the most used metrics in the classification tasks, representing the fraction of correctly classified instances given all instances.
(4)$$\;Accuracy=\frac{TP+TN}{TP+FP+TN+FN}$$
where TP, TN, FP, and FN refer to the number of True Positives, True Negatives, False Positives, and False Negatives, respectively. In such cases, high accuracy scores are relatively easy to reach, i.e., by labeling all the samples as the majority class, which, however, do not provide much value for applications. Thus, in order to assess the machine-learning models from a more comprehensive perspective, other confusion-matrix-based metrics such as Recall, macro-F1 score, and ROC AUC were introduced in this research.

Based on the confusion matrix in Table 2, precision (5), recall (6), and F1 score (7) can be calculated. In a default binary classification case, precision (P) describes the classifier’s ability to identify only the positive (“defective”) instances, recall (R) measures the capacity of identifying all the positive (“defective”) instances, while the F1 score considers both precision and recall, measuring the ability of the model to identify instances of a specific class. Finally, the macro F1 scores (8) can be calculated by averaging the F1 scores of both classes. In industrial applications such as intrusion analysis or breakout prediction, anomaly detection where the data are highly complicated and true positive instances could come at a greater cost, a classifier with a higher recall score and a certain exchange of precision could be accepted as well. Hence, in order to further examine the models’ ability to recognize normal samples and to simplify the calculation of macro and weighted F1 scores, the precision and recall of negative samples have been calculated as well, with the recall of the positive class being taken as an auxiliary metric. For a similar purpose, ROC AUC has also been introduced as an alternative metric to evaluate how well the models performed on the dataset.
(5)$$Precision=\frac{TP}{TP+FP}$$
(6)$$Recall=\frac{TP}{TP+FP}$$
(7)$${F1}_{positive}=2\times \frac{Precision\times Recall}{Precision+Recall}$$
(8)$${F1}_{macro}=\sum_{c}\frac{2}{c}\times \frac{{Precision}_{c}\times {Recall}_{c}}{{Precision}_{c}+{Recall}_{c}}$$
(9)$${F1}_{macro}=\sum_{c}2\times w_{c}\frac{{Precision}_{c}\times {Recall}_{c}}{{Precision}_{c}+{Recall}_{c}}$$

The receiver operating curve (ROC) is another evaluation tool utilized in this case study, which represents an approach to visually evaluate the performance of a given model by visualizing the ratio between TP and FP. It shows how a predictor compares with the true outcome. It is a graphical diagram demonstrating the ability to detect the correct predictions in a binary classification system. More specifically, the ROC plots the true positive rate (TPR) versus the false positive rate (FPR). Here, TPR (10) refers to the ratio of TP (correct classified “defective”) instances to the total number of Positive (“defect”)samples, while FPR (11) is the ratio of FP (incorrect predictions for the “normal” instances) to the total number of Negative (“normal”) samples. This criterion is defined as follows [40]:(10)$$\;TPR=\frac{TP}{TP+FN}$$
(11)$$\;FPR=\frac{FP}{FP+TN}$$

One of the large advantages of ROC analysis is that it is threshold-agnostic; the performance of a predictor is estimated without a specific threshold and gives a criterion to choose an optimal threshold based on a certain cost function or objective [41,42]. It can also be synthesized numerically by calculating the total area under the curve (AUC). A random classifier would get a ROC AUC score of 0.5, while a perfect classifier would make it 1.0. Figure 8 gives an example of a ROC plot.

### 3.2. Comparison between Machine-Learning Models

Next, the algorithms mentioned in Section 2.3.1, were employed using the Scikit-learn Python library. The models were first trained on the training set, optimized on the validation set using the Hyperopt optimization pack, and finally evaluated on the testing set. The maximization of recall was adopted as the optimization target, for it was considered one of the most robust metrics for imbalanced classification.

The main hyperparameters of machine-learning models, along with their tuned values in case 4-NM, are listed below:KNN: The optimum number of neighbors found was 3, and the best distance metric was determined as cosine. The weight of the points was considered by the inverse of their distance to the query instance.SVM: Two SVM classifiers, one using a linear kernel function and the other using a sigmoid kernel. The C parameter of linear SVM was determined as 23.21, and the gamma was set to 0.825, while the C and gamma for sigmoid SVM were found to be 8.67 and 0.01, respectively. Since ROC AUC was introduced as the optimization target, probability estimation was enabled.DT: The feature selection criterion was set as “gini”; the maximum depth of tree mode was limited to 50 in order to decrease overfitting.RF: The optimum number of DTs in the ensemble was found to be 200, with the maximum feature set to 80 to decrease overfitting.AdaBoost: The base estimators were chosen as DTs with max deep of 1. The optimal number of estimators was found to be 45, with the optimal number determined as “SAMME.R”, and the learning rate found to be 0.25ANN: The optimal multi-level perceptron (MLP) network consisted of 3 hidden layers with 50 nodes each. The learning rate was established as “inverse scaling”, which gradually decreases the learning rate. The “sgd” optimizer was used to train the networks.

Taking models trained on the Case 1–4 dataset obtained by the NM under-sampling technique, and the models trained on the un-resampled dataset as examples, the recall, F1 scores, and ROC AUCs are presented in Table 3. The computing times of machine-learning models are presented in Table 4.

From Table 3 and Table 4, the most significant outcomes are summarized below:Among all the machine-learning algorithms, DT and RF models generally performed better in terms of sensitivity to defective samples in the dataset, while still maintaining certain capability to correctly classify the normal instances. The RF model obtained relatively stable performances against the changes in sampling rate, indicating that the RF model is possibly the most capable among all the machine-learning algorithms investigated, given the specific dataset and training strategy.Overall, the performance of the ANN (MLP) model dropped more severely as the under-sampling technique removes the normal cases from the training set. The authors consider that the weights and biases generally take large training sets to fully converge, and thus are not suitable for under-sampling algorithm-processed datasets.Performances of all machine-learning models were lower than expected, which possibly indicates that the distinctions between normal and defective samples are minor from the feature space, and thus feature engineering procedures might have to be improved.All machine-learning algorithms except SVMs and ANN have obtained short computing times (under 0.1s) on the Case-4 NM dataset. The authors suggest that the longer computing time of SVM could be the result of the nonlinearity of feature space, which makes it harder to create optimal hyperplanes for classification.As expected, accuracy and weighted F1 scores obtained from the test set are substantially insensitive metrics for imbalanced classification tasks, which offer deceptively optimistic results. As is presented in Case 0, considering the defective rate of the raw dataset was merely 4.2%, a model labeling all samples as negative (“normal”), could obtain a high accuracy of 95.8% while providing nearly no useful information for quality prediction and knowledge discovery. For a similar reason, although ANN models have been generally more “accurate” on all the datasets, generally low recall rates could not provide much information for defect selection, and they could not satisfy the target of the case study.In such imbalanced classification cases where the F1 scores of majority classes were superb while those of minority classes were terrible, the combination of recall, macro-F1, and ROC AUC could be an optimal combination of performance assessment.Through the under-sampling of training samples, the prediction ability of the model for defective samples could be improved, while the prediction abilities for models could drop significantly, which is frequently referred to as the “precision-recall tradeoff”. Thus, considering the overall performances, Case-4-NM datasets were regarded as favorable for model training.

### 3.3. Impacts of Resampling Techniques

Another focus of this research is to select the best data-balancing (over-sampling or under-sampling) techniques for solving the impacts of data imbalance. In this subsection, several Case 4 datasets have been obtained using NM, SMOTE, RUS, and ROS, respectively, to train RF models and compare the results. The recall, F1 scores, and ROC AUCs are presented in Table 5, and the computing time of RF models on Case 4 datasets are presented in Table 6.

From Table 5, it was evident that the RF models trained on ROS and SMOTE datasets did not perform significantly better than those trained on Case 0 (unsampled) datasets. This may be because, adding new instances into datasets under highly skewed datasets could introduce excessive noise in the feature space, on which the machine-learning models may tend to overfit, and eventually lose part of the ability for classification. In comparison, the over-sampling techniques may also lead to large datasets and longer training time (although in this case, such changes in training time are considered tolerable, according to Table 6). Thus, it is possibly more favorable to adopt under-sampling for such large and complex datasets.

## 4. Conclusions and Prospects

This paper describes a comprehensive case study for the sensor data-driven inclusion prediction of the continuous-casting process. A wide range of machine-learning algorithms, including decision trees, support vector machines, K-nearest neighbors, random forests, AdaBoost, and multi-layer perceptron neural networks, have been established, trained, and examined on an actual sensor dataset consisting of mold level fluctuation, casting speed, and flow rates of mold cooling water, collected from 7300 samples, 327 of which were defective, throughout the timespan of one month. The model inputs correspond to statistical features from EMD IMFs of sensor signals, while the output of the classification of the binary labels indicates the instances of inclusion-caused rolling line lockdowns. The best results were obtained with a random forest on the Near-Miss under-sampled training set (ROC AUC: 0.64; accuracy: 0.77; recall for defective samples: 0.41).

Based on the numerical results of experiments, it is fair to conclude that, taking the limited variables and the complexity of the process, this case study has provided one of the most precise approximations of data-driven quality prediction processes for continuous-casting products. Based on existing long sensor sequences from 7000 samples and quality labels from quality-inspection systems, the machine-learning pipelines have obtained acceptable, or even promising, results. It has been proven that class imbalance could be tackled with distance-based under-sampling results, and machine-learning algorithms could deliver capable predictions. For the actual continuous-casting process, such data-driven quality predictions could help decision-makers estimate defect rates and optimize production parameters accordingly. This could reduce production and inspection costs and thus is valuable for modern intelligent steelmaking processes.

Although the research obtained certain results, there are still problems to be solved. The accuracy and recall scores remain issues to be improved. Due to the high complexity of the continuous-casting process, the accessible data in the current research may not fully represent the slag-inclusion mechanisms in the continuous-casting process. A small number of defective instances, coupled with under-sampling techniques, may also lead to insufficient information about normal samples in the training sets. Extensive research and experiments are also needed to further interpret how signal features and classification thresholds could influence the final outputs. Fortunately, emerging time-series-based signal-processing techniques such as shapelet transformation [43] and deep-learning techniques have provided alternatives for advanced feature representation and classification alternatives, on which future research could focus on. The authors also consider further research on the combination of quality prediction and scheduling tasks in casting, rolling, refining, etc.; the reference related to this can be found in [44].

## Figures and Tables

**Figure 1 sensors-23-06719-f001:**
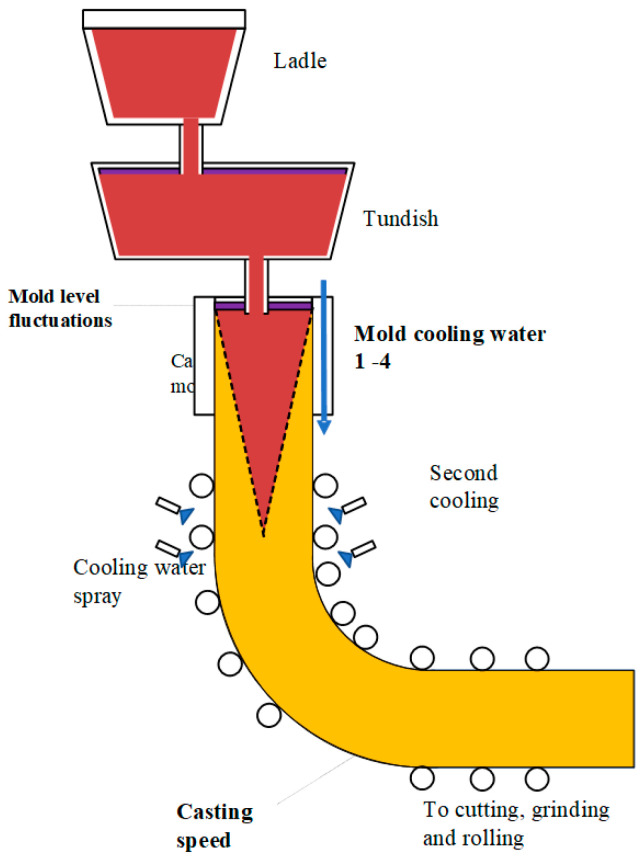
The schematic diagram of a continuous–casting machine.

**Figure 2 sensors-23-06719-f002:**
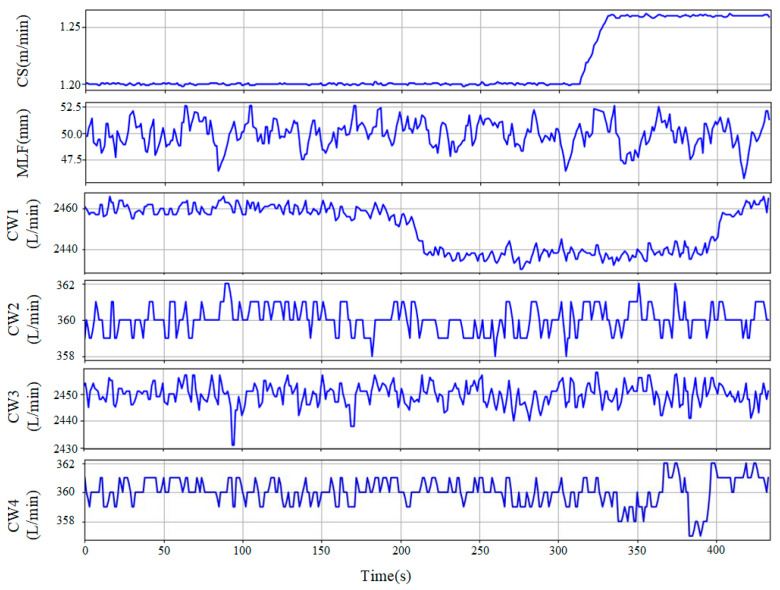
Time-series curves of continuous-casting slab (Normal sample) no. 205CS5400.

**Figure 3 sensors-23-06719-f003:**
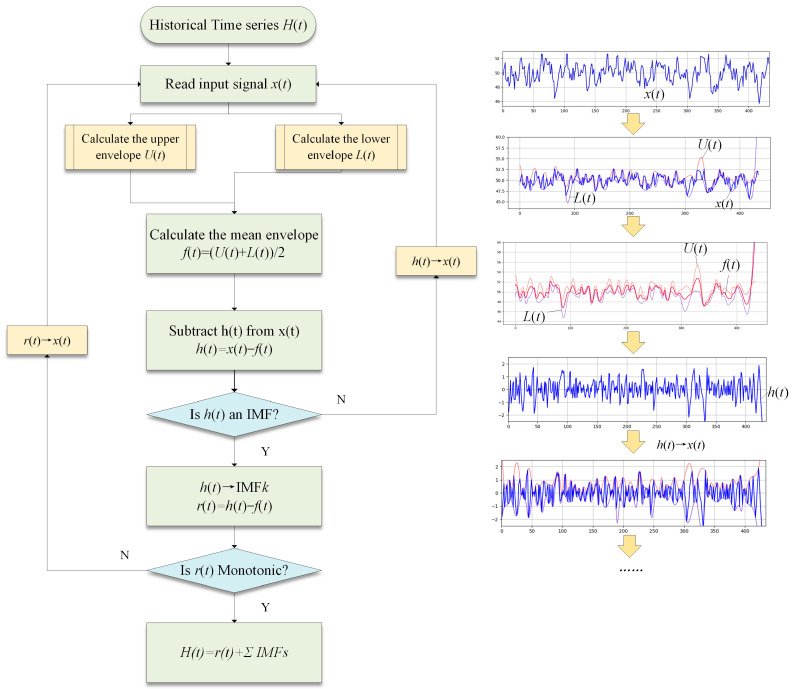
The general flowchart of EMD. A brief example of the calculation of “IMF” and “Residual” in each iteration has been placed on the right side of the figure.

**Figure 4 sensors-23-06719-f004:**
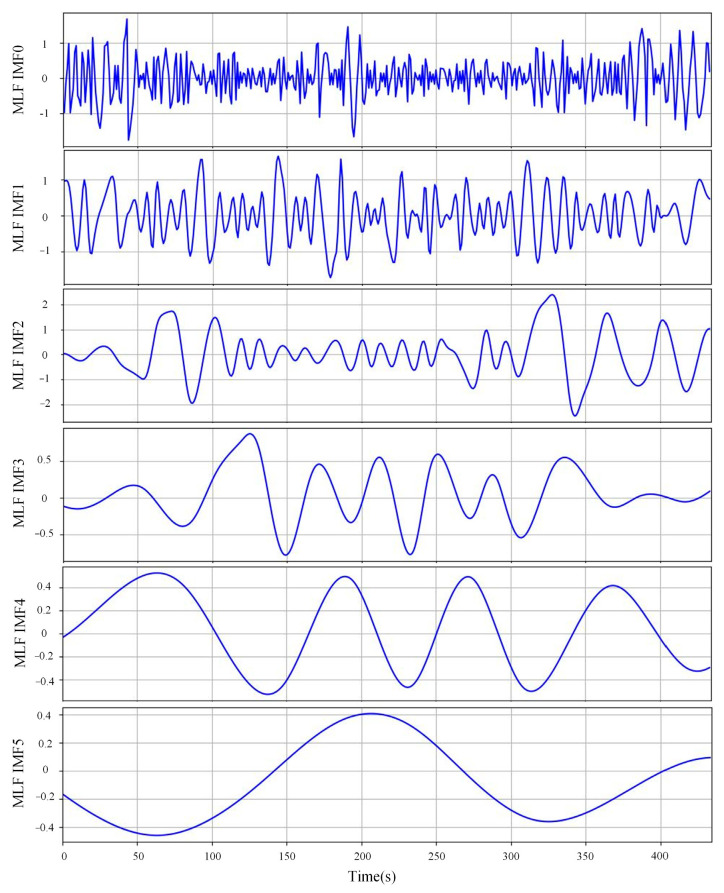
EMD decomposition results for MLF signal from slab no. 210CS5400.

**Figure 5 sensors-23-06719-f005:**
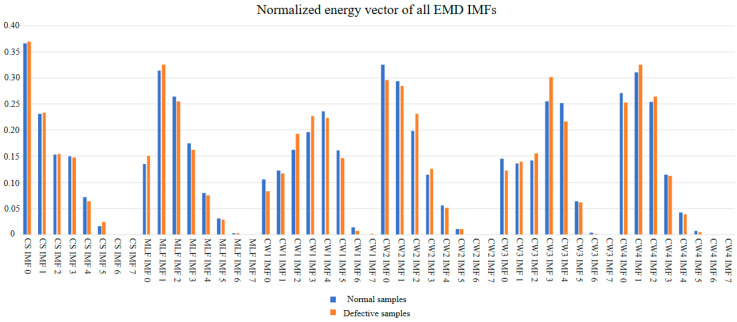
Normalized energy spectrum for all EMD IMFs.

**Figure 6 sensors-23-06719-f006:**
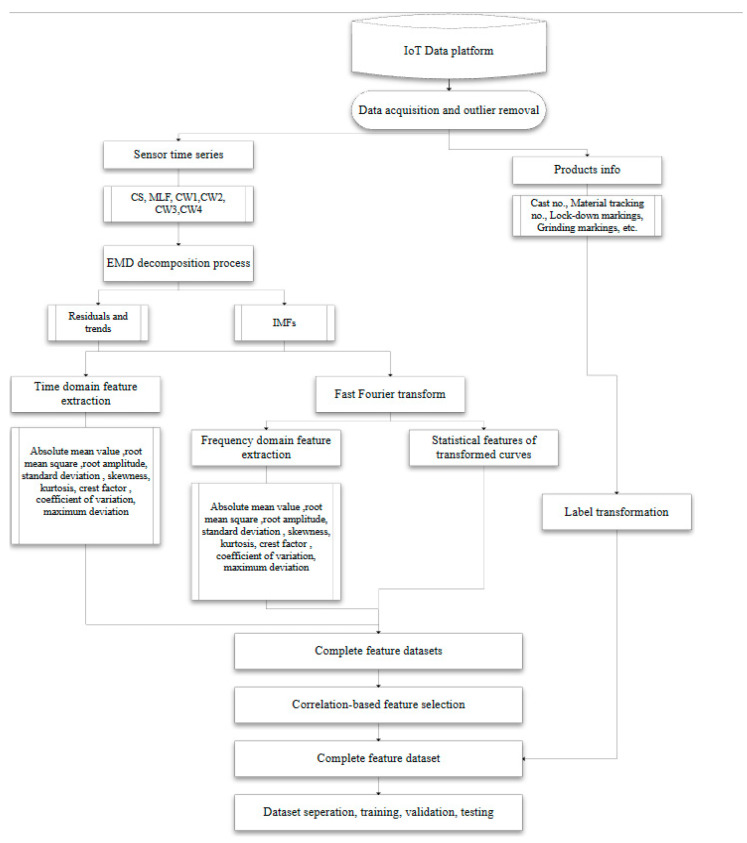
The process for data acquisition, signal processing, feature extraction, and selection.

**Figure 7 sensors-23-06719-f007:**
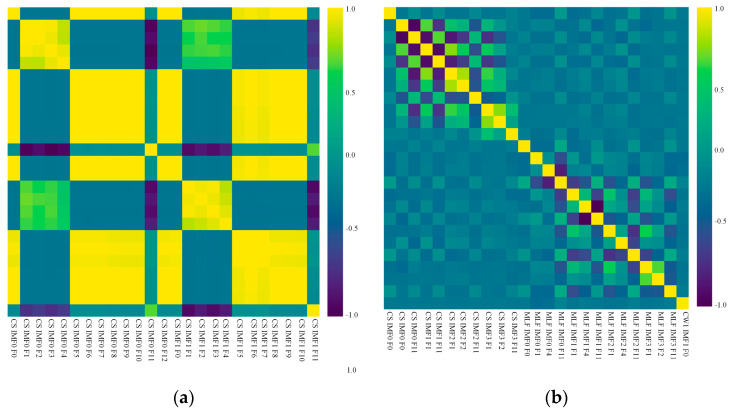
The correlation heatmap of the top 25 features before (**a**) and after (**b**) feature selection.

**Figure 8 sensors-23-06719-f008:**
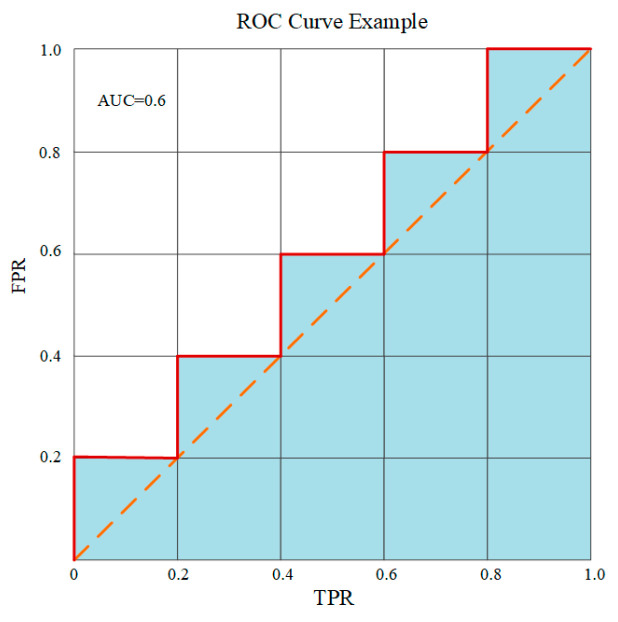
An example of the ROC curve and its AUC score.

**Table 1 sensors-23-06719-t001:** The parameter settings and outputs of 4 representative samples in the dataset. (Sample cast no. processed due to data security agreements).

Sample Cast No.	Status	Parameter Settings	Value	Parameter Output	Range
210CS5300	Defective	CS (m/min)	1.31	CS	1.21–1.31
		MLF (mm)	46	MLF	44.8–47.5
		CW1 (L/min)	2520	CW1	2504–2530
		CW2 (L/min)	360	CW2	357–362
		CW3 (L/min)	2520	CW3	2503–2531
		CW4 (L/min)	360	CW4	357–363
205CS5400	Normal	CS (m/min)	1.2	CS	1.198–1.262
		MLF (mm)	50	MLF	45.7–52.2
		CW1 (L/min)	2450	CW1	2430–2466
		CW2 (L/min)	360	CW2	358–362
		CW3 (L/min)	2450	CW3	2431–2458
		CW4 (L/min)	360	CW4	357–362
210CS0200	Normal	CS (m/min)	1.2	CS	1.012–1.263
		MLF (mm)	90	MLF	87.4–93.4
		CW1 (L/min)	400	CW1	389–407
		CW2 (L/min)	1950	CW2	1945–1956
		CW3 (L/min)	400	CW3	398.7–401.8
		CW4 (L/min)	1950	CW4	1947–1956.2
210CS5100	Defective	CS (m/min)	1.22	CS	1.128–1.221
		MLF (mm)	46	MLF	44.3–48.2
		CW1 (L/min)	2520	CW1	2512–2540
		CW2 (L/min)	360	CW2	358–362
		CW3 (L/min)	2520	CW3	2509–2530
		CW4 (L/min)	360	CW4	354–363

**Table 2 sensors-23-06719-t002:** Confusion matrix of actual class and predicted class for binary classification.

		Actual Label	
		0	1
**Predicted label**	0	TN	FP
	1	FN	TP

**Table 3 sensors-23-06719-t003:** Performances of machine-learning algorithms in all cases using the NM technique.

Case	Model	Recall: Normal	Recall: Defective	Macro-F1	Weighted-F1	ROC AUC	Accuracy
Case 4-NM	RF	0.76	0.41	0.53	0.83	0.64	0.77
	DT	0.68	0.42	0.45	0.70	0.57	0.67
	SVM-linear	0.78	0.25	0.47	0.82	0.60	0.75
	SVM-sigmoid	0.79	0.27	0.49	0.80	0.55	0.77
	ANN	0.98	0.11	0.56	0.93	0.63	0.94
	KNN	0.86	0.19	0.50	0.83	0.54	0.83
	AdaBoost	0.98	0.11	0.56	0.93	0.63	0.94
Case 3-NM	RF	0.71	0.42	0.47	0.79	0.62	0.70
	DT	0.66	0.58	0.45	0.63	0.60	0.65
	SVM-linear	0.72	0.27	0.45	0.76	0.51	0.70
	SVM-sigmoid	0.82	0.28	0.51	0.87	0.61	0.82
	ANN	0.89	0.18	0.51	0.86	0.54	0.86
	KNN	0.69	0.46	0.46	0.76	0.57	0.68
	AdaBoost	0.96	0.11	0.54	0.92	0.60	0.92
Case 2-NM	RF	0.55	0.59	0.4	0.67	0.62	0.55
	DT	0.53	0.58	0.36	0.58	0.55	0.53
	SVM-linear	0.59	0.43	0.45	0.63	0.51	0.58
	SVM-sigmoid	0.79	0.22	0.49	0.83	0.55	0.77
	ANN	0.48	0.46	0.36	0.51	0.52	0.49
	KNN	0.57	0.42	0.40	0.67	0.53	0.55
	AdaBoost	0.88	0.16	0.5	0.88	0.58	0.85
Case 1-NM	RF	0.32	0.83	0.28	0.44	0.58	0.32
	DT	0.41	0.7	0.34	0.52	0.57	0.39
	SVM-linear	0.48	0.61	0.33	0.56	0.56	0.49
	SVM-sigmoid	0.50	0.48	0.36	0.63	0.51	0.49
	ANN	0.28	0.87	0.22	0.39	0.52	0.34
	KNN	0.62	0.42	0.34	0.57	0.55	0.43
	AdaBoost	0.58	0.44	0.4	0.7	0.54	0.58
Case 0	RF	1	0	0.49	0.93	0.65	0.96
	DT	1	0	0.49	0.93	0.63	0.96
	SVM-linear	0.96	0.08	0.52	0.92	0.51	0.92
	SVM-sigmoid	0.96	0.06	0.50	0.92	0.50	0.92
	ANN	1	0	0.49	0.93	0.55	0.96
	KNN	1	0	0.49	0.93	0.52	0.96
	AdaBoost	0.95	0.12	0.53	0.91	0.61	0.91

**Table 4 sensors-23-06719-t004:** Performances of machine-learning algorithms in case 4 using the NM technique.

Cases	Models	Computing Time
Case 4-NM	RF	0.087 s
	DT	0.039 s
	SVM-linear	8.957 s
	SVM-sigmoid	0.596 s
	ANN	0.263 s
	KNN	0.075 s
	AdaBoost	0.069 s

**Table 5 sensors-23-06719-t005:** Performances of RF models on Case 4 datasets.

Cases	Models	Recall: Normal	Recall: Defective	Macro-F1	Weighted-F1	ROC AUC	Accuracy
Case 4-NM	RF	0.76	0.41	0.53	0.83	0.64	0.77
Case 4-RUS		0.68	0.47	0.45	0.77	0.61	0.67
Case 4-ROS		0.99	0	0.49	0.93	0.63	0.93
Case 4-SMOTE		0.94	0.01	0.50	0.92	0.65	0.94

**Table 6 sensors-23-06719-t006:** Computing times of RF models on Case 4 datasets.

Cases	Model	Computing Time
Case 4-NM	RF	0.087 s
Case 4-RUS		0.089 s
Case 4-ROS		0.093 s
Case 4-SMOTE		0.091 s

## Data Availability

The data presented in this study are available on request from the corresponding author. The data are not publicly available, because the data is a part of one ongoing research project, where, according to the data management agreement with 3rd party, data sharing is restricted for authorized usage only.

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
