# Peer review of "Machine-Learning Algorithms for Process Condition Data-Based Inclusion Prediction in Continuous-Casting Process: A Case Study"

_sensors, 2023, doi:10.3390/s23156719_

Round 1
Reviewer 1 Report
In this case study, a machine learning algorithm was developed to predict slag inclusion defects in continuous casting slabs, based on process condition sensor data. The following machine learning algorithms have been examined: K-Nearest neighbors, support vector classifier (linear and nonlinear kernels), decision trees, random forests, AdaBoost and Artificial Neural Networks. The topic is interesting, and the logic is complete. But some improvements can be conducted:
1. Some of the contents are not very clear, i.e., what is the full name of “CL1” in Table 1? What is the difference between “MLF” and “MLE” in Table 1?
2. The picture is not clear enough, such as Figure 3. The formats of Eq. 2 and Eq. 3 are incorrect.
3. Table 3 lists performances of different machine learning algorithms. But what is the run time of different machine learning algorithms?
4. In Table 3, the accuracy of ANN and KNN also reach 0.96 for Case 0, but why did you choose optimized random forest in Table 4?
No.
Author Response
We sincerely thank the reviewer for the valuable feedback that we have used to improvethe quality of our manuscript. The reviewer comments are laid out below in italicized font and specific concern shave been numbered. The authors' response is given in normal font. Revisions made according to reviewer comments have been marked with yellow highlight.
Point 1:
Some of the contents are not very clear, i.e., what is the full name of “CL1” in Table 1? What is the difference between “MLF” and “MLE” in Table 1?
Response:
The "CLs" and "MLE" in Table 1 was due to writing errors in manuscript preparation. The “CL1-4” and “MLE” in Table 1 should have been “CW1-4” and "MLF" described in 2.2, respectively:
"In this case study, 6 process parameter which were considered as key variables by expert on-site, were analyzed: cast speed (CS, m/min,), mold level fluctuation (MLF, mm), flow rates of mold cooling water from the four sides of casting mold (CW 1-4, L/min)."

Abbreviations and markings in other parts of the manuscripts have also been checked.
Point 2:
The picture is not clear enough, such as Figure 3. The formats of Eq. 2 and Eq. 3 are incorrect.
Response:
All the figures have been rearranged with Visio in higher resolutions. For instance, Fig.3 has been rearranged as follows:

Formats of all equations have been checked and revised using Math Editor.
Point 3:
Table 3 lists performances of different machine learning algorithms. But what is the run time of different machine learning algorithms?
Response:
Run time consumptions have been recorded in the initial experiments. In the revised version, run times have been added to the manuscript as Table 4 and Table 6. Table 4 included run times of all models on Case 4-NM dataset, and Table 6 included run times of RF model on all Case 4 datasets.
Discussions regarding computing time have also been added in subsection 3.2:
"
3...
4. All machine learning algorithms except SVMs and ANN have obtained short computing times (under 0.1s) on the Case-4 NM dataset. The authors suggest that the longer computing time of SVM could be the result of nonlinearity of feature space, which makes it harder to create optimal hyperplanes for classification.
5....."
Point 4:
In Table 3, the accuracy of ANN and KNN also reach 0.96 for Case 0, but why did you choose optimized random forest in Table 4?
Response:
To choose an optimal classification algorithm for imbalanced dataset, just looking at accuracy might not be enough. As has been stated in 3.1:
“high accuracy scores are relatively easy to reach, i.e., by labeling all the samples as the majority class, which, however, do not provide much value for applications.”
Although KNN and ANN have obtained the accuracy of 96%, their zero recall rates indicated that they failed to recognize any of the defective samples in the dataset. In comparison, random forest classifiers have obtained generally higher recall rates and overall accuracies, which means they could recognize more defective samples, while retaining acceptable accuracies (In subsection 3.2):
"Through the under-sampling of training samples, the prediction ability of the model for defective samples could be improved, while the prediction abilities for models could drop significantly, which is frequently referred to as the ‘Precision-Recall tradeoff’. Thus, considering the overall performances, Case-4-NM datasets were regarded favorable for model training."
To highlight the reason for not selecting ANN classifiers for defect prediction, a more detailed explaination has been added in subsection 3.2:
"
4....
5. As expected, accuracy and weighted F1 scores obtained from the test set are substantially insensitive metrics for imbalanced classification tasks, which offer deceptively optimistic results. As is presented in Case 0, Considering the defective rate of the raw dataset was merely 4.2%, a model labeling all samples as negative (“Normal”), could obtain a high accuracy of 95.8% while providing nearly no useful information for quality prediction and knowledge discovery. For the similar reason, although ANN models have been generally more “accurate” on all the datasets, generally low recall rates could not provide much information for defect selection, they could not satisfy the target of the case study.
6. ...."
Reviewer 2 Report
Dear Authors,
I have attached the review report. Please examine it.
Best regards,

Moderate editing of the English language is required. The authors should have used a grammar check editor for some grammatical errors and misprints.
Author Response
The authors sincerely thank the reviewer for the valuable feedback that we have used to improvethe quality of our manuscript. The reviewer comments are laid out below in italicized font and specific concern shave been numbered. The authors' response is given in normal font. Revisions made according to reviewer comments have been marked with green highlight.
Point 1:
The authors should have used a grammar check editor for some grammatical errors and misprints.
Response:
The authors have used Grammarly check for the whole paper during revision, and the grammatical errors and misprints have been corrected. For example:
In introduction:
"… More specifically, modelling and analysis with regards to the steel cleanliness, which implies a tight control of the inclusion formation, is essential to fulfil the increasing demands for quality and efficiency."
In the revised version:
"More specifically, modelling and analysis concerning steel cleanliness, which implies a tight control of the inclusion formation, is essential to fulfill the increasing demands for quality and efficiency."
Point 2:
The title
Machine learning algorithms for process condition data-based inclusion prediction in continuous casting process: a case study
should be:
Machine learning algorithms for process condition data-based inclusion prediction in continuous casting process: A case study
Response:
The title has been changed to:
“Machine learning algorithms for process condition data-based inclusion prediction in continuous casting process: A case study”.
Point 3:
They should mention the following state-of-the-art classifiers in Introduction Section:
EigenClass - A Precise and Stable Machine Learning Algorithm: Eigenvalue Classification (EigenClass), Neural Computing and Applications, 33(10), 5381 5392 (2021). https://doi.org/10.1007/s00521-020-05343-2
Multinomial Naive Bayesian Classifier - Multinomial Naive Bayesian Classifier Framework for Systematic Analysis of Smart IoT Devices Adaptive Two-Index Fusion Attribute-Weighted Naive Bayes, Sensors 2022, 22(19), 7318; https://doi.org/10.3390/s22197318
FPFS- NN - Fuzzy Parameterized Fuzzy Soft -Nearest Neighbor Classifier, Neurocomputing, 500, 351 378 (2022). https://doi.org/10.1016/j.neucom.2022.05.041
FR-NN - Fuzzy Rough Nearest Neighbour Methods for Aspect-Based Sentiment Analysis, Electronics 2023, 12(5), 1088; https://doi.org/10.3390/electronics12051088
Response:
The state-of-the-art classifiers have been mentioned and cited in the Introduction Section in a new paragraph as Ref. [23], Ref. [22], Ref. [20], and Ref. [21], respectively.
"
....Similarly, Wu et al. [17] adopted multi-scale CNN and RNN models for reliable quality prediction of continuous casting slabs. The authors also conducted data balancing based on the random under-sampling (RUS) method to mitigate the impact of the skewed data distribution.
Apart from quality inspection in the continuous casting process, other state-of-the-art research has also revealed the potential of machine learning algorithms in providing pre-cise and interpretable predictions in various applications. Dhaliwal et al. [18] proposed an XGBoost-based model for measuring the parameters of data in networks. Zhang [19] pro-posed cost-sensitive K-nearest neighbor (KNN) classifiers by minimizing misclassification costs in imbalanced classification processes. Memiş et al. [20] combined soft sets and KNN classifiers, forming a fuzzy parameterized fuzzy soft KNN classifier, which obtained optimal performances in a comprehensive comparison experiment. Also based on fuzzy theories, Kaminska et al. [21] proposed an improved KNN using ordered weighted average operators and obtained optimal results in applied aspect-based sentiment analysis. Kaushik et al. [22] used a multinomial naive Bayesian Classifier for text classifica-tion-based sentiment analysis. Erkan [23] reported an eigenvalue-based algorithm, using the eigenvalues of each sample in the test data with concerning the training data to make classifications.
The studies mentioned above, as well as others, have shown the potential of data-driven machine learning methods in analyze and estimation of production processes in intelligent manufacturing.
To the best of authors’ knowledge...
"
Point 4:
Mathematical expressions should be rearranged by using Math Editor.
Response:
Formats of all equations have been rearranged using Math Editor.
For example, in 3.1. "Evaluation metrics":

Point 5:
The figures should be changed with their high-resolution versions.
Response:
All figures have been rearranged with Visio in higher resolutions. For example, Figure 2 in the revised manuscript:

Point 6:
Table 2 should be rearranged by deleting excess spaces.
Response:
Table 2 has been rearranged:
Point 7:
In binary classification, precision and recall are determined based on positive as defaults. Please check their expression in the paper.
Response:
The description of precision and recall has been altered. Related paragraph in subsection 3.1 has been modified as:
"Based on the confusion matrix in Table 2, Precision (5), Recall (6), F1 score (7) can be calculated. In a default binary classification case, precision (P) describes the classifier’s ability to identify only the positive (“Defective”) instances, recall (R) measures the capacity of identifying all the positive (“Defective”) instances, while F1 score considers both precision and recall, measuring the ability of the model to identify instances of a specific class. Finally, the macro F1 scores (8) can be calculated by averaging the F1 scores of both classes...Hence, in order to further examine the models’ abilities to recognize normal samples, and to simplify the calculation of macro and weighted F1 scores, Precision and Recall of negative samples have been calculated as well, with Recall of positive class being taken as an auxiliary metric. For the similar purpose, ROC AUC has also been introduced as an alternative metric to evaluate how well the models performed on the dataset."
Point 8:
Conclusion section mentions disadvantages, limitations of the study, and the future works. Conclusion section should be detailed by mentioning advantages of the paper.
Response:
The contributions of the case study have been summarized in subsection 4 “Conclusions and prospects”:
"
...the Near-Miss under-sampled training set (ROC AUC: 0.64; Accuracy: 0.77; Recall for defective samples: 0.41.).
Based on the numerical results of experiments, it is fair to conclude that, taking the limited variables and the complexity of the process, this case study has provided this paper has provided one of the most precise approximation of data-driven quality prediction processes for continuous casting products. Based on existing long sensor sequences from 7000 samples and quality labels from quality inspection systems, the machine learning pipelines have obtained acceptable, or even promising, results. It has been proved that class imbalance could be tackled with distance-based under-sampling results, and machine learning algorithms could deliver capable predictions. For actual continuous casting process, such data-driven quality predictions could help decision makers to estimate defect rates, and optimize production parameters accordingly. This could reduce production and inspection costs, and thus is valuable for modern intelligent steelmaking processes.
Although the research obtained certain results, there are still problems to be solved. The accuracy and recall scores remain as issues to be improved....
"
Reviewer 3 Report
This work describes a comprehensive case study for the sensor data-driven inclusion prediction of continuous casting process. Six machine learning techniques including decision trees, support vector machines, K-nearest neighbors, random forests, AdaBoost and multi-layer perceptron neural networks, have been established, trained, and examined on an actual sensor dataset.
Some more detailed comments are given below. I hope that if the authors will take them into account the paper will be improved.
1. Place table title of table 1 above the table.
2. There are many state-of-the-art softwares, including Rapidminer, KNIME, KEEL, Weka, Tanagra, Orange, can perform the sensor data-driven inclusion prediction. Authors should survey and discuss these state-of-the-art softwares in Section 2.
3. Authors should compare the performance of six approaches with state-of-the-art softwares, including Rapidminer, KNIME, KEEL, Weka, Tanagra, Orange, etc. in Section 4.
4. Please ensure that every reference cited in the text is also present in the reference list (and vice versa). This article does not cite the Ref. [25] in the text.
5. It is hard to link the EMD with the actual sensor dataset in Section 3. A simple example is needed to provide an illustration of the flowchart of EMD in Figure 3.
Author Response
We sincerely thank the reviewer for the valuable feedback that we have used to improvethe quality of our manuscript. The reviewer comments are laid out below in italicized font and specific concern shave been numbered. The authors' response is given in normal font. Revisions made according to reviewer comments have been marked with blue highlight.
Point 1:
Place table title of table 1 above the table.
Response:
The position of table title has been changed.

Point 2:
There are many state-of-the-art softwares, including Rapidminer, KNIME, KEEL, Weka, Tanagra, Orange, can perform the sensor data-driven inclusion prediction. Authors should survey and discuss these state-of-the-art softwares in Section 2.
Response:
The authors agree that more studies would be useful to understand details of machine learning algorithms and their implementations. But, the authors have found that, such discussions and comparisons are hard to make in this specific research paper, since these are platforms to integrate common algorithm models, rather than independent algorithms for classification or prediction. For instance, In "Feature extraction using wavelets and classification through decision tree algorithm for fault diagnosis of mono-block centrifugal pump" (Measurement 46.1 (2013): 353-359, doi:https://doi.org/10.1016/j.measurement.2012.07.007), the authors used C 4.5 decision tree in Weka for fault diagnosis, which was similar to CART decision tree using entropy criterion in the Scikit-learn library (https://scikit-learn.org/stable/modules/tree.html).
However, the description of these machine learning toolboxes, as well as the reason for choosing Scikit-learn for case study, have been mentioned in 2.4.1:
"In recent years, various machine learning packages and toolboxes have been released to handle the complexity and high dimensionality in actual industrial scenarios. Packages such as Scikit-learn, Spark ML-Lib, and Weka have provided highly compact and flexible solutions for building up custom analysis workflow within programming environments, while tools like KEEL, RapidMiner, Tanagra, and Orange provided nearly end-to-end software applications for developing machine learning algorithms. Considering package compatibility, deployment complexity and overall dependability (Junaid, M.; Ali, S.; Siddiqui, I.F.; Nam, C.; Qureshi, N.M.F.; Kim, J.; Shin, D.R. Performance Evaluation of Data-Driven Intelligent Algorithms for Big Data Ecosystem. Wireless Pers Commun 2022, 126, 2403–2423, doi:10.1007/s11277-021-09362-7.) , the authors have selected Scikit-learn, a Python machine learning package under constant development and with good algorithm support, to configure the machine learning models. The pipeline also used the libraries Numpy and Pandas for reading data inputs, and Matplotlib and Seaborn for visualization and analysis for results. After the data was allocated, preprocessing and feature extraction tasks were carried out, and the selected features were used as input for classifiers. The following classification algorithms have been used in this research: ...."
Point 3:
Authors should compare the performance of six approaches with state-of-the-art softwares, including Rapidminer, KNIME, KEEL, Weka, Tanagra, Orange, etc. in Section 4.
Response:
As Ref. [30] in the revised version has pointed out, actual outputs and performances could vary considering programming approaches of implementation, and the authors are interested in examining such differences. However, such comparison between multiple algorithm implementations in multiple packages/ toolboxes requires another set of comprehensive studies and experiments which, the authors are afraid, are away from the focus of the study, i.e., to examine the best algorithm for quality prediction of casting products, rather than to compare algorithm instances in different packages. The authors plan to investigate machine learning packages and platforms in another independent case study, in order to discuss such issues from a boarder perspective.
Point 4:
Please ensure that every reference cited in the text is also present in the reference list (and vice versa). This article does not cite the Ref. [25] in the text.
Response:
The reference list and citations have been checked by the authors. The Ref. [25] (Chorowski, J.; Wang, J.; Zurada, J.M. Review and Performance Comparison of SVM- and ELM-Based Classifiers) has been cited as Ref. [32] in 2.4.1, line 320, p.10:
"Support Vector Machines (SVM): SVM is a supervised learning method widely applied in classification and regression tasks. By mapping the data into high-dimensional space with kernel functions and constructing proper hyperplanes [32], SVM could effectively handle various classification datasets, even when the data distributions are nonlinear [33]...."
Point 5:
It is hard to link the EMD with the actual sensor dataset in Section 3. A simple example is needed to provide an illustration of the flowchart of EMD in Figure 3.
Response:
The Figure 3 has been updated with an example on how the envelopes and “residuals” are calculated in each iteration:

Round 2
Reviewer 2 Report
Dear Authors,
In the revised manuscript, many suggestions have been adopted, and major corrections have been made. Therefore, my evaluation is that the revised version of this manuscript can be published in this journal. However, the resolution of the figures and mathematical expressions should be checked before the publication process.
Best regards,
The authors have used a grammar check editor for some grammatical errors and misprints. Therefore, minor editing of the English language is required.
Reviewer 3 Report
The authors have carefully addressed the previous comments of the reviewer and significantly improved the manuscript.